# Diode Laser Management of Primary Extranasopharyngeal Angiofibroma Presenting as Maxillary Epulis: Report of a Case and Literature Review

**DOI:** 10.3390/healthcare9010033

**Published:** 2021-01-01

**Authors:** Saverio Capodiferro, Luisa Limongelli, Silvia D’Agostino, Angela Tempesta, Marco Dolci, Eugenio Maiorano, Gianfranco Favia

**Affiliations:** 1Department of Interdisciplinary Medicine, University of Bari Aldo Moro, 70121 Bari, Italy; capodiferro.saverio@gmail.com (S.C.); luisannalimongelli@gmail.com (L.L.); angelatempesta1989@gmail.com (A.T.); gianfranco.favia@uniba.it (G.F.); 2Department of Medical, Oral and Biotechnological Sciences, University of Chieti Pescara, 66100 Chieti, Italy; marco.dolci@unich.it; 3Department of Emergency and Organ Transplantation, University of Bari Aldo Moro, 70121 Bari, Italy; eugenio.maiorano@uniba.it

**Keywords:** nasopharyngeal angiofibroma, extranasopharyngeal angiofibroma, diode laser, trans-mucosal photocoagulation, oral cavity

## Abstract

Juvenile nasopharyngeal angiofibroma is a rare vascular neoplasm, mostly occurring in adolescent males, and representing 0.05% of all head and neck tumors. Nevertheless, it is usually recognized as the most common benign mesenchymal neoplasm of the nasopharynx. Usually, it originates from the posterolateral wall of the nasopharynx and, although histologically benign, classically shows a locally aggressive behavior with bone destruction as well as spreading through natural foramina and/or fissures to the nasopharynx, nasal and paranasal cavities, spheno-palatine foramen, infratemporal fossa and, very rarely, to the cranial cavity. Extranasopharyngeal angiofibroma is considered a distinct entity due to older age at presentation, different localizations (outside the nasopharyngeal pterygopalatine fossa) and attenuated clinical course. Extranasopharyngeal angiofibroma has been sporadically described in the oral cavity. We report a case of extranasopharyngeal angiofibroma with primary and exclusive involvement of the adherent gingiva of the anterior maxilla, managed by preoperative diode laser trans-mucosal photocoagulation and subsequent surgical removal. The current literature on primary extranasopharyngeal angiofibroma is also reviewed.

## 1. Introduction

Juvenile nasopharyngeal angiofibroma (JNA) is an uncommon vascular tumor, mainly occurring in the nasal and paranasal cavities of adolescent males and representing approximately 0.05% of all tumors of the head and neck [1,2,3,4]. Morphologically, JNA would fit the criteria for benign mesenchymal neoplasms but may show a locally aggressive clinical course in view of local extension into the adjacent tissues, which often precludes complete surgical removal, with possible tumor persistence and recurrences in a relatively high number of patients (20%) [4,5]. This represents a true challenge for JNA management, as such tumors frequently arise in the posterior-lateral wall of the nasal cavity close to the superior margin of the sphenopalatine foramen and may cause bone erosion and displacement of adjacent structures [5,6,7,8,9,10]. There is accumulating evidence that a morphologically similar angiofibroma, namely extranasopharyngeal angiofibroma (ENA), should be considered as a distinct subtype. In fact, it is generally characterized by occurrence in a wider age range, localization outside the nasopharyngeal pterygopalatine fossa, frequent involvement of the nasal septum and the maxillary sinus, and usually shows a more attenuated clinical course, with low propensity for recurrence [11].

Intraoral localization of ENA has been occasionally reported in the lip, palate, tonsil, tonsillar pillar, cheek, retro-molar area and gingiva [12,13,14,15,16,17,18,19,20,21,22,23,24,25,26,27,28,29,30,31]. We report on a case of extranasopharyngeal angiofibroma (ENA) with localization to the adherent maxillary gingiva and without maxillary sinus involvement. The diagnostic work-up and the surgical management by preoperative diode laser transmucosal photocoagulation, and subsequent removal of the lesion by conventional scalpel surgery, are accurately described.

In addition, all previously reported cases of ENAs were selected from the literature, collected and discussed. All angiofibromas arising from or involving the nasal/maxillary cavities were excluded, as well as cases with primary manifestation in the cheek as an extension of a tumor arising elsewhere, in order to better define the clinico-pathological features of ENA with exclusive localization in the oral cavity.

## 2. Case Presentation

The patient was a 17-year-old Caucasian male, referred to our attention for persistent gingival swelling. His medical history was unremarkable, with the exception of massive bleeding during an attempt to surgically remove the same gingival lesion performed by his general practitioner two months earlier. The patient referred no previous bleeding at other sites and his blood tests were within normal limits.

At intra-oral examination, a soft, firm and painless swelling of the adherent gingiva (between 2.3 and 2.4 teeth) of red-violet discoloration was detected (Figure 1). Radiological examination showed an irregular radiolucency between the aforementioned dental roots with periodontal enlargement (Figure 2a) and cortical bone erosion (Figure 2b), with irregular margins (Figure 2c). Both teeth responded positively to the vitality tests.

Based on anamnestic data, pre-photocoagulation by diode laser, aimed at reducing the intra-lesion vascular component, with a simultaneous punch-biopsy, were suggested before surgical removal. After local anesthesia, a transmucosal photocoagulation was performed by diode laser (GaA1As-A2G laser “Surgery35”, A&G s.r.l., Italy) with a wavelength of 800 ± 10 nm, a 320 µm flexible fiber and an energy output of 5 W in continuous emission modality. The procedure lasted approximately 20 s and resulted in color variation from red-violet to white-greyish. A small punch biopsy was performed followed by histological examination of the sample. On hematoxylin-eosin stained slides, the tumor was composed by spindle-shaped fibroblast-like cells immersed in a rather loose collagenous stroma with vascular lacunae containing coagulated and fragmented red blood cells, sometimes showing a ghost-like appearance. On these bases, the diagnosis of angiofibroma was rendered. (Figure 3)

One month later, the lesion appeared slightly reduced in size and reddish in color (Figure 4a), with persistent inflammation of the marginal gingiva, thus leading to an adjunctive diode laser treatment with the same modalities (Figure 4b). After another month, inflammatory signs were almost absent and excisional surgery was performed as follows: elevation of a para-marginal periosteal flap, scalpel removal of the intra-osseous lesion (Figure 4c), accurate curettage of the adjacent bone and dental roots. A final histological examination confirmed the original diagnosis on punch biopsy. Capillary-sized blood vessels showed an irregular and curvilinear profile and intraluminal thrombosis. Occasionally, some vessels appeared wider and lined by a thicker and more intensely collagenized wall. Overall, both fibroblast-like and endothelial cells did not show relevant nuclear atypia or mitotic activity (Figure 5a–c). Such morphologic features were also detectable in the intraosseous component of the lesion and were consistent with the final diagnosis of ENA.

The patient was monitored until complete mucosal healing (Figure 6a) and followed-up for three years with no recurrence. (Figure 6b).

This study was carried out in accordance with the code of ethics of the world medical association (Declaration of Helsinki), approved by internal ethical committee (study number 4575, prot. 1442/C.e). The patient released informed consent on diagnostic and therapeutic procedures and for the possible use of biological samples for research purposes.

## 3. Discussion

The first description of a vascular tumor with prominent fibrous stroma most probably dates back to Hippocrates in the 5th century B.C., while Chauveau in 1906 first used the designation juvenile nasopharyngeal angiofibroma [32]. In 1983, Fisch stratified JNA in four stages, mainly based on radiological findings [33]. Such stratification was subsequently revised by Andrews and Fisch in 1989 who distinguished JNA into four types: Type I = tumor limited to the nasopharyngeal cavity with negligible bone destruction or limited to the sphenopalatine foramen; Type II = tumor invading the pterygopalatine fossa or the maxillary, ethmoid or sphenoid sinuses with bone destruction; Type III = tumor invading the infratemporal fossa or the orbital region without intracranial involvement (a) or with intracranial extradural (parasellar) involvement (b); Type IV = intracranial intradural tumor without (a) or with (b) infiltration of the cavernous sinus, pituitary fossa, or optic chiasm [33]. Despite further attempts of classification, as reported by Alshaikh et al. in an overview in 2015 [3], what really matters about JNA is the lack of clinical signs and symptoms at the early stages of the disease. In fact, JNAs are occasionally detected as nasopharyngeal masses on radiograms taken for other medical problems, or manifest unspecific signs such as unilateral nasal obstruction, recurrent epistaxis, etc. [6,9,34]. At more advanced stages, patients may show facial swelling, cranial neuropathy, proptosis and limitation of mouth opening [1,2,3,5,6,7,33,34]. In addition, a biopsy is generally contraindicated in such instances because of the risk of potentially fatal bleeding. Consequently, the diagnosis essentially is based on matching clinical and radiological (CT, MRI, angiography) findings [6,7,9,10,35].

Conversely, JNAs with maxillary involvement, with or without extension into the oral cavity, are more easily diagnosable, and an incisional biopsy is performable in most cases. Nevertheless, the differential diagnosis remains extremely challenging, especially for the presence of undefined radiological borders, thus including benign/malignant epithelial and odontogenic neoplasms, benign and malignant maxillary/nasal sinuses neoplasms, localizations of hematological malignancies and metastatic tumors [6,36,37,38,39,40,41].

Extranasopharyngeal localizations of angiofibroma are widely documented in the literature, [2,4,8,11] although there is accumulating evidence that ENA should be considered a distinct entity in view of older age at onset, predilection for females and different localization and clinical presentation. ENA also displays a poorer blood vessel component and more intense fibrosis at histological examination, and less propensity for recurrence after surgical treatment. [2,3,13,23,36,42,43].

In a detailed review published in 2018 by Windfuhr et al., [11] 174 cases of ENA were collected and reviewed. The most frequently involved sites were the nasal septum (39 cases—22.4%), maxilla (23 cases—13.2%), and inferior turbinate (20 cases—11.5%), followed by ethmoid (13 cases—7,5%), upper aerodigestive tract (including hypopharynx, larynx, trachea and oesophagus) (12 cases—6.9%), nasal cavity (12 cases—6.9%), oropharynx (nine cases—5.2%), oral cavity (nine cases—5.2%), sphenoid sinus (seven cases—4%), cheek (five cases—2.9%). Such data confirm the propensity of ENAs to occur in the nasal cavities (especially the nasal septum), maxilla, ethmoid, larynx and also the possible involvement of areas at distance from the original site of origin. This may frequently be the case for angiofibromas involving the cheek and manifesting as facial swellings but taking their origin in the nasal/maxillary cavities or in the pharynx. ENAs involving the maxilla, with or without gingival manifestations, usually are extensions of a tumor arising in the nasal or maxillary cavities [36,37,42] In this regard, in 2005 our group published the case of an angiofibroma with primary intra-oral presentation [37] localized in the posterior maxillary gingiva of a young girl, but with involvement (and most probable origin) in the maxillary sinus. Therefore, we concur with Windfuhr et al., who stated that “It would be ideal to differentiate histologically and demographically between intranasal and other locations of extranasopharyngeal angiofibromas (ENA) when analysing these tumours in the future. ….. In this current review of the literature, however, it was impossible to make this distinction based on the given information in the analysed literature.” [11]

According to this statement, in the current review of the literature, we exclusively selected cases with primary oral involvement, paying attention to attentively identify those ENAs occurring in the maxilla (or maxillary gingiva/palatal mucosa) that showed no involvement of the nasal/maxillary cavities. To the best of our knowledge and with the exclusion of the current case, a total of 20 cases of ENA matching the aforementioned criteria have been previously reported [12,13,14,15,16,17,18,19,20,21,22,23,24,25,26,27,28,29,30,31], the pertinent data of which on age, sex, site, symptoms and author/year of publication are listed in Table 1.

Overall, there were 15 males and five females, the age range was 0–60 years old in males and 14–87 years old in females. Two ENAs occurred in newborns. More precisely, one case was reported as congenital and the other occurred at the age of six months. A higher prevalence (eight cases) was detected among adolescent males, with an age range of 10–23 years old. As to localizations, four cases occurred in the buccal mucosa, four in the mandible, five in the tonsil/tonsillar pillar, one in the soft palate, one in the lip, three in the hard palate/premaxilla, and two in the cheek; the latter were intramural lesions without nasal cavity involvement, as confirmed by radiograms. [14,21] Foreign body sensation, chewing and speech difficulties were usually related to site and size of the lesions, while bleeding was referred or observed during clinical/surgical procedures in seven cases only. In two cases the lesions were treated by laser [15,27].

The case reported herein show overlapping clinical features with previously reported ENAs, as occurring in an adolescent male (17 years old), showing bleeding at the first attempt of surgical removal and representing a diagnostic challenge based on its clinic-radiological presentation. In fact, the peripheral localization, along with the radiological appearance, pointed at a wide range of clinical diagnoses including several reactive or neoplastic lesions with gingival/periodontal onset (such as periodontal infections, proliferative osteitis, exostosis, peripheral osteoma, peripheral ossifying fibroma, giant cell granuloma, and lobular capillary hemangioma) [13,36,37,38,42,44,45,46].

As to treatment options, it is generally accepted that the introduction of laser therapies in the last three decades has simplified oral surgery and avoiding or consistently reducing bleeding during laser incision of soft tissues, promoting intralesional coagulation in vascular tumors [47,48,49,50]. Therefore, lasers with proven surgical capabilities are nowadays widely adopted to treat benign, premalignant and malignant neoplasms, nonsurgical periodontal diseases and small and large (infiltrative) vascular malformations [47,48,49,50,51,52,53]. The photocoagulative effects of lasers such as Diode, Neodimio:YAG, KTP, CO2, are strictly related to the intrinsic property of the laser light, at specific wavelengths, to interact with oxyhemoglobin, thus inducing photothermolysis, erythrocyte microagglutination and blood vessel obliteration [49,51,53]. In the current case, the occurrence of intraoperatory bleeding at the first surgical approach, as well the clinic-radiological appearance of the lesion, pointed at a possible diagnosis of a richly vascularized lesion, which would have probably taken advantage of presurgical transmucosal photocoagulation. Such an initial therapeutic approach, without curative intents, in consideration of the somewhat worrisome radiologic presentation, would have also allowed incisional biopsy to possibly better define the nature of the nodule. Notably, the sample obtained by punch biopsy, although showing slight photocoagulative alterations as a consequence of laser irradiation, allowed proper identification of the tumor and to rule out the many other neoplasms that would have required distinct surgical options.

Overall, the diagnosis of JNA/ENA generally is achieved by conventional histological examination, being characterized by a dense fibrous stroma, containing spindled to stellate stromal cells, and haphazardly arranged vascular channels of variable size [2,3,5,7,8,35,37,54]. Vimentin, CD34 and α-actin immunostains may support the diagnosis in equivocal cases.

The etiology of JNA and ENA is still debated and unclear. With regard to JNA, different theories have been postulated but a general agreement is still missing. JNA has been alternatively interpreted as a hamartoma [40,55,56,57,58], a vascular malformation (related to incomplete regression of the first branchial arch artery) [28], overgrowth of paraganglionic tissue [40] or extracolonic manifestation of familial adenomatous polyposis. [2,4,8,10,11,40,55,57,58] To date, according to the most accepted theory [8,9,10,11,56,57,58,59], JNA may be the consequence of repeated microhemorrhages followed by repair by granulation tissue and subsequent fibrous tissue deposition, and possibly due to sexual hormone stimulation of the vascular erectile tissues located in the proximity of the sphenopalatine foramen.

As an extrapolation of this theory, we may assume that ENA may possibly derive from ectopic (erectile) vascular tissue proliferation within the incisive (premaxillary) bone suture, with subsequent local permeation of the periodontum and, finally, vestibular gingival outgrowth, the latter being possibly related with recurrent gingival inflammation.

While considering that several studies are still ongoing to better define the overall pathogenesis of angiofibromas [59,60,61], we would like to stress that additional reports are also needed to further characterize ENA and, possibly, to better define the reasons for its relatively more indolent clinical behavior in comparison with JNA, in apparent opposition to similar invasive properties of adjacent tissues, including bones, as also illustrated in the current case.

## Figures and Tables

**Figure 1 healthcare-09-00033-f001:**
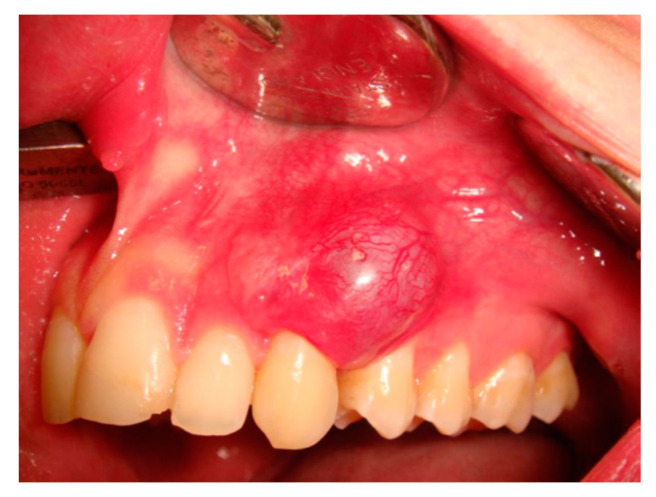
Soft red-violet swelling of the adherent gingiva in a 17 y.o. male patient.

**Figure 2 healthcare-09-00033-f002:**
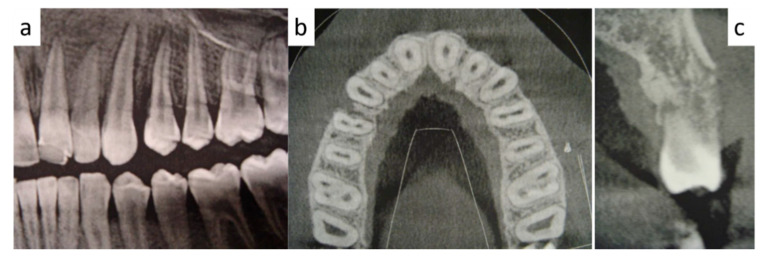
(**a**) Panoramic radiogram showing an irregular radiolucency between the dental roots of 2.3 and 2.4 teeth, with periodontal enlargement; on CT scans, vestibular cortical erosion (**b**) and irregular and margins were detected(**c**).

**Figure 3 healthcare-09-00033-f003:**
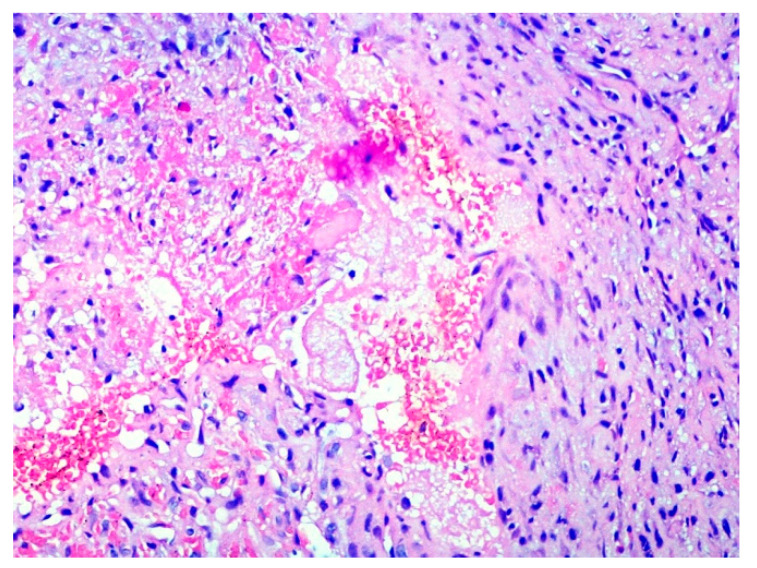
Fibrous connective stroma containing abundant vascular tissue; vascular lacunae appeared filled by fragmented and coagulated red blood cells, sometimes with a ghost-like appearance, as a result of the diode laser photocoagulation.

**Figure 4 healthcare-09-00033-f004:**
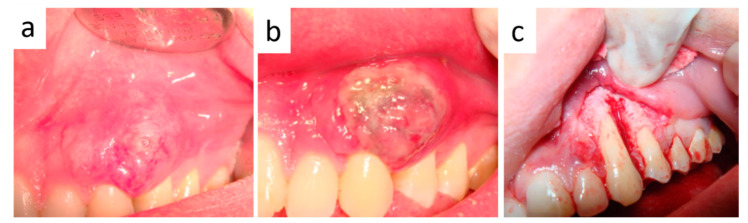
(**a**) Clinical appearance after diode laser trans-mucosal photocoagulation: the lesion appears light reddish in color and reduced in size. (**b**) Clinical appearance immediately after the second diode laser treatment: the lesion shows grey-yellowish discoloration. (**c**) Intra-operative view after surgical removal: bone erosion with regular bleeding is evident, mimicking a periodontal defect.

**Figure 5 healthcare-09-00033-f005:**
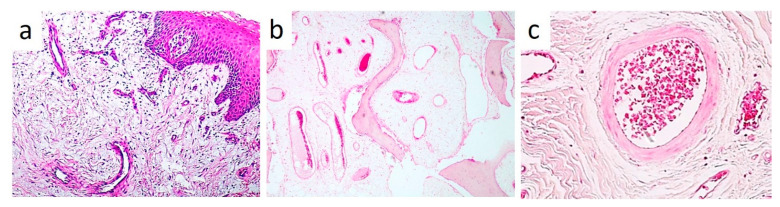
(**a–c**) Histological examination showing a rather loose fibroblastic proliferation containing several irregularly shaped blood vessels located just below the gingival epithelium (**a**) (Haematoxylin & Eosin, X4). The neoplastic proliferation also involves the adjacent bone (**b**) (H&E, X4). At higher power view, most vessels are of capillary size but some of them appear wider and with a thicker collagenized wall (**c**) (H&E, x10).

**Figure 6 healthcare-09-00033-f006:**
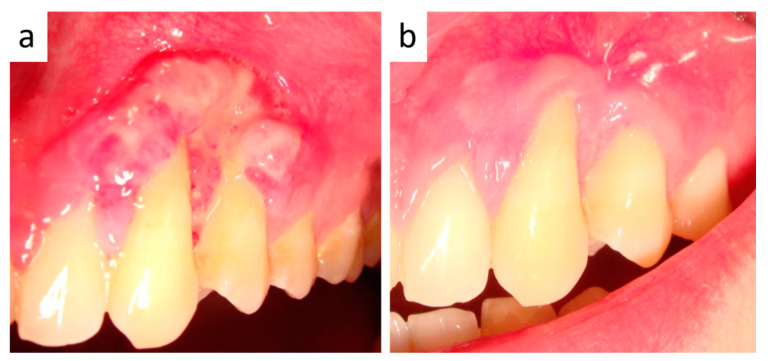
(**a**) Partially healed gingiva 12 days after surgery. (**b**) The patient was followed-up for three years with no recurrences.

**Table 1 healthcare-09-00033-t001:** Clinical features of extranasopharyngeal angiofibromas (ENAs) of the oral cavity reported in the international literature.

CASE	AUTHOR/YEAR	AGE	SEX	SITE	MAIN SYMPTOMS
1	Amini-Salari/2020	23	M	Mandibular ramus (intraosseous)	None
2	Goud/2018	30	M	Upper lip	Previous trauma, swelling
3	Jeong/2017	50	F	Cheek (intramural)	Swelling
4	Thakur/2014	87	F	Buccal mucosa	Difficulties in chewing
5	Szymańska/2013	49	M	Palatine tonsil	Dysphagia
6	Singh/2013	51	M	Buccal mucosa	Difficulties in chewing
7	Mendoza-Ramirez/2012	60	M	Tonsil	Dysphagia
8	Bakhshi/2011	50	M	Buccal mucosa	Bleeding and pain
9	Eftekharian/2008	19	M	Posterior tonsillar pillar	Foreign body sensation
10	Dere/2006	17	M	Cheek	Swelling
11	Celik/2005	15	M	Tonsil	Dysphagia
12	Andreadis/2004	6 months	M	Hard palate	Unavailable
13	Antoniades/2002	14	M	Hard palate	Difficulties in chewing/speech, bleeding on mastication
14	Kintarak/1999	46	F	buccal mucosa	None
15	Chung/1995	21	M	Soft palate	None
16	Manjalay/1992	0-congenital	M	Premaxilla	Swelling and bleeding during biopsy
17	Supiyaphun/1986	14	M	Mandible	Bleeding, impaired teeth closure
18	Ali/1982	28	F	Tonsil	Foreign body sensation, occasional bleeding
19	Reddy/1979	14	F	Mandible/retromolar area	Swelling and bleeding
20	Stewart/1973	10	M	Oral cavity/retromolar	Bleeding

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
