# Peer review of "Diode Laser Management of Primary Extranasopharyngeal Angiofibroma Presenting as Maxillary Epulis: Report of a Case and Literature Review"

_healthcare, 2021, doi:10.3390/healthcare9010033_

Round 1

Reviewer 1 Report

This study by Capodiferro et al. reports a case of adolescent angiofiroma in the gingiva. Overall, while this case report addresses the clinical course of primary gingival angiofibroma with diode laser management, the main conclusion of the report is not supported by clear, convincing evidences. The concerns are outlined in the comments below.

Major comments:

1. The juvenile nasopharyngeal angiofibroma (JNA) was so-called because it was considered as originating from the nasopharynx. Currently, the pterygoid canal is considered as the origin of JNA, therefore JNA could invade the sphenopalatine foramen, sphenoid sinus and pterygopalatine fossa. The term, extranasopharyngeal angiofibroma (ENA) has been used for angiofibromas developed in extranasopharyngeal areas with the maxillary sinus being the most commonly involved site. However, the etiology of ENA is totally different from JNA and have to be acknowledged as a different entity. ENA has virtually nothing in common with JNA and the comparison between the present case and JNA may therefore be confusing and misleading. It should be compared to other ENA cases.

2. Because the ENA in the oral cavity is extremely rare, literature review should be summarized in table and the comparative knowledge between them should be deeply discussed in the Discussion section. More detailed information about demographics, location, size, and postoperative clinical course of the oral ENA cases would be beneficial to the readers, and I would like to know more definitive findings as oral ENA in the present tumor compared to the other reported oral ENA cases.

3. Definitive diagnosis is not supported by sufficient data. More detailed clinical and histopathological evidences as characteristic features of ENA are needed in the Case Presentation section.

4. Differential diagnosis like hemangioma or angiomatos epulis should be discussed in the Discussion section. Oral hemangiomas are usually seen on the gingiva and these lesions often appear to arise from the interdental gingival papilla. It is not clear how the authors diagnosed the present case as ENA.

5. As authors pointed out, differential diagnosis as malignant neoplasms exist. Bone erosion and irregular cortex margin were seen in the CT image. It is not clear how authors excluded the possibility of malignancy and justified the use of laser treatment in the present case.

6. In the histopathological examination in Figure 4, I would like to see more histopathological findings of the residual tumor (e.g. vascular malformation, fibrous lesions, etc) after photocoagulation.

7. The conclusion of the report is not supported by clear, convincing evidences.

Minor comments:

1. Please rephrase the Figure 1 legend "referring an attempt of surgical removal failed for uncontrollable intraoperative bleeding". Case report legend should describe the key clinicopathological findings that are seen in the figure.

2. In Figure 2, please add arrows to clearly indicate the lesions so that readers can immediately find the points. Is the CT image flipped horizontally?

3. What days after photocoagulation was the Figure 3a picture taken? Please include the timeline information in the legend. I would suggest to arrange the pictures according to the timeline.

Reviewer 2 Report

The authors reported a case of extranasopharyngeal juvenile angiofibroma presenting as a maxillary epulis. As the title states that the study has performed a Literature review, the authors did not report or perform review of the previously reported cases of primary extranasopharyngeal juvenile angiofibroma. Either the title should be changed or clinicoradiographic features such as age, sex, location, clinical appearance, radiographic features, etc. of reported cases should be presented.

Many sentences were oddly structured, and presented with grammatical concerns. A few examples:

- Frequent is the involvement of the nasopharynx………

- Common finding is the absence of peripheral demarcation and permeation of the surrounding tissues, thus explaining the tendency of Jas to persist of recur.

- One similar case of PEJA of the gingiva we found by searching in literature reported by ……

The authors should perform extensive English editing.

Round 2

Reviewer 1 Report

The various suggestions made were well addressed.

Reviewer 2 Report

There is considerable improvement in the manuscript after revision.